# Tuning direct-written terahertz metadevices with organic mixed ion-electron conductors

Cristiano Bortolotti[1,2,9], Federico Grandi [3,4,9], Matteo Butti[2], Lorenzo Gatto [3,4,8], Francesco Modena[2], Christina Kousseff[5], Iain McCulloch [5,6], Caterina Vozzi [4], Mario Caironi[2], Eugenio Cinquanta [4] ✉ & Giorgio Ernesto Bonacchini [7] ✉

In the past decade, organic mixed ion-electron conductors have been successfully adopted in innovative bioelectronic, neuromorphic, and electro-optical technologies, as well as in multiple energy harvesting and printed electronics applications. However, despite the intense research efforts devoted to these materials, organic mixed conductors have not yet found application in electronic/photonic devices operating in key regions of the electromagnetic spectrum, such as the microwave (>5 GHz) and terahertz (0.1-10 THz) ranges. A possible reason for this technological gap is the widespread notion that organic electronic materials are unsuitable for high-frequency applications. In this work, we demonstrate for the first time the utility of high-performance polymer mixed conductors as electro-active tuning layers in reconfigurable terahertz metasurfaces, achieving modulation performances comparable with state-of-the-art inorganic and 2D semiconductors. Through time-domain terahertz spectroscopy, we show that the large conductivity modulations of these polymers, until now probed only at very low frequencies, are effectively preserved in the terahertz range, leading to optimal metadevice reconfigurability. Finally, we leverage the unique processability of organic materials to develop fully direct-written electrically tuneable metasurfaces onto both rigid and flexible substrates, opening new opportunities for the mass-scale realization of flexible and light-weight terahertz optics with unique mechanical characteristics and environmental footprint.

In the past two decades, the quest for low-cost and portable terahertz (THz) technologies has been the object of intense interdisciplinary efforts, given the critical role that electronic and hybrid electronic–photonic systems operating in this frequency range will likely play in upcoming sensing, imaging, and telecom applications[1,2].

These efforts have delivered a wealth of THz technological platforms based on inorganic semiconductors (e.g. III-V compounds and silicon), 2D materials (e.g. graphene and transition metal dichalcogenides), superconductors, metal oxides (e.g. vanadium dioxide) and liquid crystals[3,4]. Each of these material strategies presents relative

[1]Dipartimento di Elettronica, Informazione e Bioingegneria, Politecnico di Milano, Milan, Italy. [2]Center for Nano Science and Technology, Istituto Italiano di Tecnologia, Milan, Italy. [3]Dipartimento di Fisica, Politecnico di Milano, Milan, Italy. [4]Istituto di Fotonica e Nanotecnologie, Consiglio Nazionale delle Ricerche, Milan, Italy. [5]Department of Chemistry, University of Oxford, Oxford, UK. [6]Andlinger Center for Energy and the Environment and Department of Electrical and Computer Engineering, Princeton University, Princeton, USA. [7]Department of Information Engineering, Università degli Studi di Padova, Padua, Italy. [8]Present address: Max-Planck-Institut für Quantenoptik, Garching, Germany. [9]These authors contributed equally: Cristiano Bortolotti, Federico Grandi. ✉e-mail: eugenioluigi.cinquanta@cnr.it; giorgio.bonacchini@unipd.it

advantages, as well as unique trade-offs between device performances and ease of implementation, and all have been successfully integrated with metallic metasurface constructs to boost both their functionalities and performance[5]. Indeed, the ability to combine metallic resonant structures with actively tuneable materials (e.g. semiconductors) has led to several reconfigurable metasurface concepts also known as metadevices[6] whose electromagnetic properties can be dynamically controlled through either thermal, optical, or electrical stimuli. Despite the applicability of this general principle to many different combinations of active materials and passive metallic metasurfaces, no THz metadevice based on organic electronic materials has so far been proposed[7–9]. Interestingly, very few fundamental works investigate the THz properties of electrically-modulated organic (semi)conductors[10], with none focusing on new high-performance mixed ion-electron conductors[11,12], which have recently enabled significant advances in bioelectronic[13], neuromorphic[14], and electro-optical technologies[15], as well as multiple energy harvesting[16] and printed electronics[17] applications. A possible explanation of this gap in both scientific literature and technology is likely due to the relatively low charge carrier mobilities and poor charge injection of organic semiconductors with respect to other families of semiconductors, which result in relatively low device switching speeds[18]. This limitation is likely the reason for the scarce use of organic materials in high-frequency applications[19,20].

In this work, we explore for the first time the use of organic electronic materials as tuning layers within reconfigurable THz metaoptics. Indeed, we show how organic mixed ion-electron conductors (OMIECs) can effectively modulate the transmission of metallic metasurfaces resonating in the 0.5–0.75 THz range through a charge carrier screening effect, as previously demonstrated with inorganic semiconductors and layered nanomaterials[21,22]. We take advantage of an emerging class of OMIECs characterized by conjugated polymer backbones that exhibit high hole (or electron) mobilities, while also displaying liquid-like ionic conduction thanks to their hydrophilic glycolated sidechains[23]. Specifically, we employ the polymer p(g2T-TT)–the prototype compound for this class of OMIECs[24]–which we deposit by inkjet printing using a commercial printer. Thanks to this approach, we are able to fabricate THz metadevices where both the metasurface and the active materials are processed solely by cost-effective and mass-scalable direct-writing techniques, namely inkjet printing and femtosecond laser writing, which are also compatible with large-area and plastic substrates[25]. These methods offer a maskless process flow that ensures an efficient use of materials, minimizing waste. Combined with the low temperature processing allowed by OMIECs, this fabrication scheme relies on a reduced energy consumption and relatively low capital expenditure, thus contributing to its sustainability[26,27]. At the same time, the absence of critical raw materials is a further advantage[28].

## Results

### Design and operation of the metadevice

The device achitecture explored in this work is shown in Fig. 1, and consists of an inverse metasurface design[29]–also known as complementary metasurface–of 60 ×60 inverse split-ring resonators with a 50 μm lattice parameter, and a lateral gate electrode, both made of gold. On rigid fused silica substrates, this structure displays a frequency selective transmission window at approximately $f_{res,O} = 0.74$ THz for planar waves with normal incidence, and with the electric field polarized orthogonal to the gap region–see Supplementary Fig. 1 for details on the geometry and equivalent circuit model of the inverse split-ring resonators. The transmission properties of this passive metasurface were modulated via the addition of the electrically tuneable p(g2T-TT), which was deposited on the gate electrode and metasurface, and then covered with an iongel electrolyte based on 1-ethyl-3-methylimidazolium bis(trifluoromethylsulfonyl)imide (EMIM:TFSI) as gating medium. The addition of these two layers decreases both the resonance frequency, which sets at $f_{res} = 0.64$ THz (see Fig. 1b), and the base transmission of the metasurface. At this stage, all layers were deposited and patterned using conventional microfabrication techniques, such as photolithography, thermal evaporation, and spin-coating.

Complex transmission spectra on this device were acquired using THz Time-Domain Spectroscopy (THz-TDS) at different gating voltages in the 0.6 to −0.8 V range. The highest potential corresponds to a fully discharged OMIEC, and thus to the maximally transmissive state of the metasurface. As the OMIEC is gradually charged with increasingly more negative $V_{gate}$ potentials, a hole-mediated charge screening effect takes place, effectively quenching the resonant behaviour of the metasurface[9,24]. By relying on sub-volt polarization, the device exhibits both excellent amplitude and phase modulation depths–i.e. MD and $\Delta\varphi$, respectively, where the former is calculated as:

$$MD = \frac{T_{MAX} - T_{min}}{T_{MAX}} \sim 65\%$$

With $T_{MAX}$ being the transmission at $f_{res}$–i.e. the peak transmission at 0.6 V–and $T_{min}$ the transmission at the same frequency for the −0.8 V spectrum[30]. The phase modulation $\Delta\varphi$ is obtained by using the 0.6 V spectrum as reference for the other curves and reaches a maximum of approximately 60° at 0.85 THz.

### Evaluation of terahertz charge transport dynamics in the polymer

Notably, the extent of these modulations is comparable, or sometimes superior, to what is currently achieved with state-of-the-art technologies that rely solely on electrical modulation[31], with the important

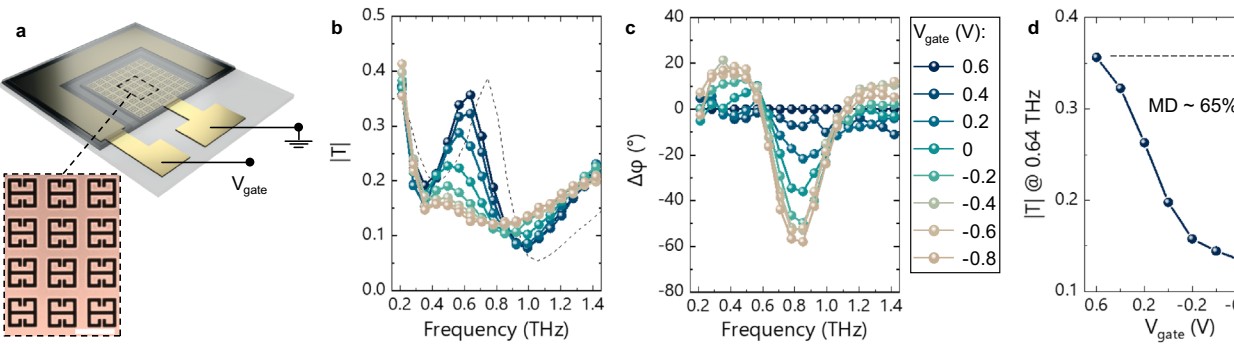

**Fig. 1 | Device structure and operation. a** Schematic representation of a OMIEC-based metadevice, with microscope image of the bare metasurface (scale bar: 50 μm). **b, c** Amplitude and phase modulation of the complex THz transmission, where the magnitude (**b**) is calculated by using a measurement without metadevice as a reference, while for the phase (**c**) the reference is instead the measurement at 0.6 V. The dashed line in **b** corresponds to the transmission of the metasurface without OMIEC and iongel. **d** Voltage dependence of the transmission at -0.64 THz.

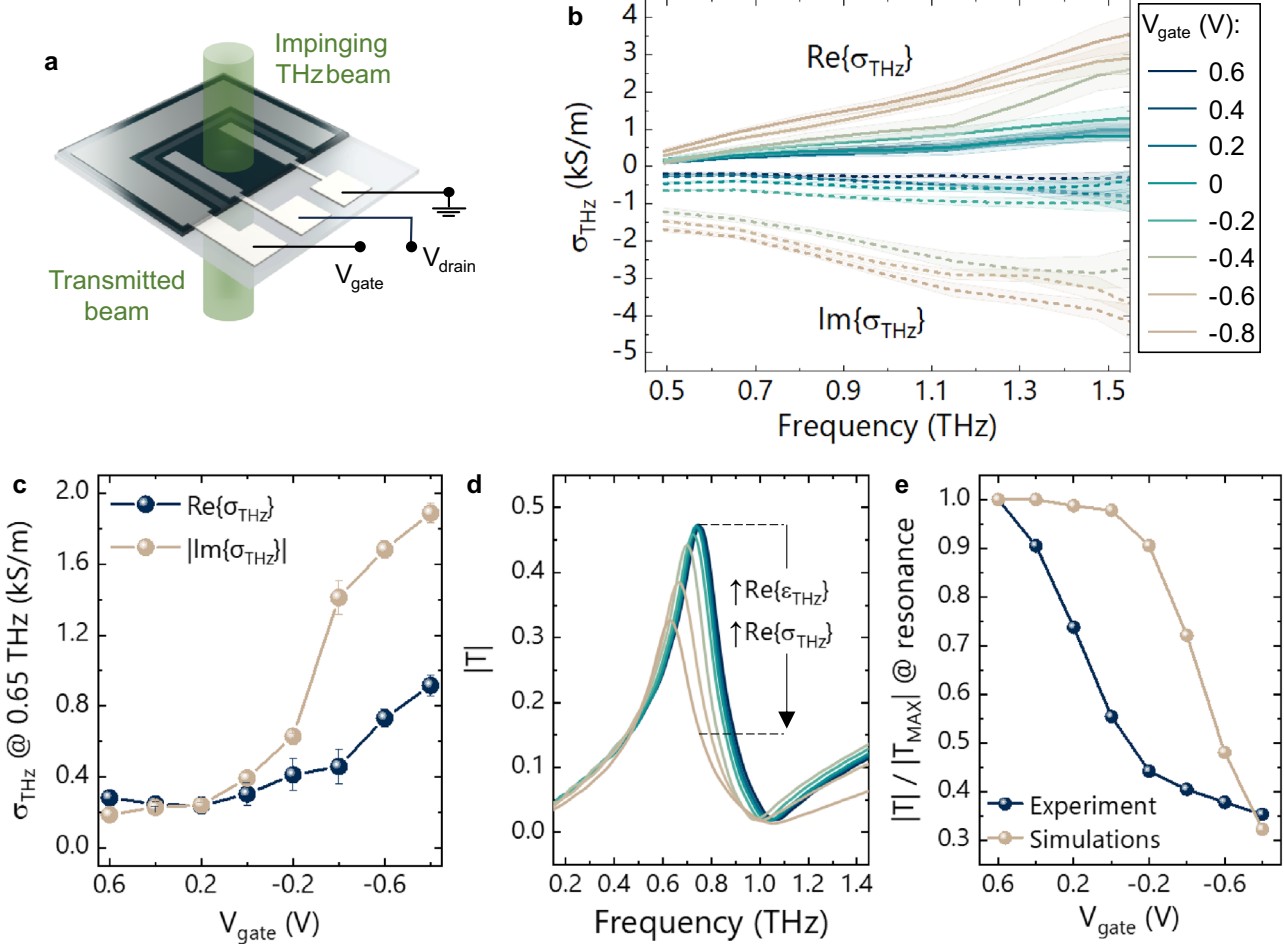

**Fig. 2 | THz conductivity of p(g2T-TT). a** Representation of the three-electrode device used to perform both electrical and THz-TDS measurements on the bare OMIEC. **b** Complex conductivity spectra of p(g2T-TT) at different gate biases (shaded areas correspond to the SE with a sample of 5 measurements). **c** Voltage dependence of the real and imaginary parts of $\sigma_{THz}$ (error bars correspond to the SE with a sample of 5 measurements). **d** Simulated transmission of a tuneable meta-surface, where the real permittivity and the THz conductivity are varied within the ranges obtained from the THz-TDS measurements. **e** Comparison of the normalised transmission at resonance for the simulated and the fabricated devices, showing similar modulation depths.

exception of the work by Zhang et al. (*MD* = 85%, *Δφ* = 68°), which is however based on much more complex and expensive materials and fabrication processes[30]. Moreover, this approach requires lower gating biases compared to other technologies, which typically need polarization voltages that are one or two orders of magnitude greater. Considering the relatively low charge carrier mobility of organic electronic materials compared to inorganic semiconductors[32], this result is rather intriguing, and for this reason, a more in-depth understanding of their charge transport properties at high frequencies is warranted. We hence analysed how both the dc and THz conductivities of p(g2T-TT) affect the behaviour of our metadevices by combining a set of electro-optical measurements with 3d electro-magnetic simulations of our architecture.

The schematics reported in Fig. 2a display the three-terminal device that was used to perform both electrical and THz-TDS measurements on bare charge-modulated p(g2T-TT) films without the underlying metasurface. With this device architecture, the three terminals can either be connected and biased as the gate, drain, and source electrodes of a conventional organic electrochemical transistor (OECT)[33], or operated as a two-electrode system by grounding both the drain and source contacts−i.e. $V_{drain} = V_{source} = 0$ V. The first config-uration allows the extraction of the steady-state electrical conductivity $\sigma_{DC}$ of the OMIEC from the characteristic transfer curve of the OECT, shown in Supplementary Fig. 2, while through the latter we perform in situ spectroscopic measurements of the THz conductivity $\sigma_{THz}$ at

different electrostatic doping levels, thus capacitively charging the OMIEC without obstructing the THz beam.

Figure 2b shows the frequency-dependent real and imaginary components of $\sigma_{THz}$ at different gate voltages, which were calculated from the THz refractive index and absorption coefficient obtained from the transmission spectra (see Methods). We first note that, as expected, the absolute values of both real and imaginary parts of $\sigma_{THz}$ increase as holes are injected into the bulk of the OMIEC layer. We then observe that the spectral evolution of the THz conductivity can be interpreted according to the Drude-Smith conduction model, which phenomenologically describes the charge transport in disordered media−including (semi)conducting polymers[10,34]−where electron (or holes) experience backscattering and confinement effects due for example to grain boundaries, finite polymer chains, or other forms of structural and energetic disorder[35,36]. The Drude-Smith model is expressed by the following equation:

$$\sigma_{THz} = \frac{\varepsilon_0 \omega_P^2 \tau}{1 - i\omega\tau} \left(1 + \frac{c}{1 - i\omega\tau}\right) \qquad (1)$$

Where $\omega_P$ is the plasma frequency, $\varepsilon_O$ the free space permittivity, $\tau$ the scattering time. The so-called localization parameter $c$ can take values comprised between 0 and −1, where with $c = -1$ the model reproduces a weakly confined system with fully localized carriers due to back-scattering, while for $c = 0$ the Drude conductivity is recovered.

As shown in Supplementary Fig. 3, the fitting of our experimental data with this model yields localization parameters that range between −0.85 and −0.98, which is consistent with previous studies performed on other organic (semi)conductors[10]. While the microscopic mechanisms leading to a partial localization of the charge in (semi)conducting polymers have not yet been fully clarified, these results confirm the relatively confined nature of polarons in these nanocomposite systems[11,34].

To correlate the THz conductivity data to the modulation characteristics of the metadevices, we analysed the gate voltage dependence of $\sigma_{THz}$ at the frequency of operation of the metasurface (i.e. 0.65–0.75 THz). We hence note that the minimum and maximum values−i.e. the conductivities obtained at 0.6 and −0.8 V, respectively −fall in the 245–915 S/m range for the real part, and between the 185–1890 S/m for the imaginary one (see Fig. 2c). These conductivity modulations are inferior to those obtained in dc from the electrical characterisation of p(g2T-TT)-based OECTs, which typically span more than three orders of magnitude in the 0.1–$10^4$ S/m range (see Supplementary Fig. 2)[24]. At low $V_{gate}$ polarizations, the relatively high $\sigma_{THz}$ values can be explained by considering that the THz beam effectively probes all charge carriers within the OMIEC, including the ionic species that passively diffuse throughout the bulk of the polymer when placed in contact with the ionic liquid[23]. Therefore, while not effectively contributing to the electrically measured conductivity $\sigma_{DC}$, it is likely that these ionic charges−possibly along with other deeply trapped states−establish the $\sigma_{THz}$ baseline at low gating voltages. On the other hand, at high hole concentrations, both the real and imaginary parts of $\sigma_{THz}$ fall short of the dc conductivity by a factor of 4 and 2, respectively. We hence performed parametric simulations of the metadevice by simultaneously modifying the real parts of both the OMIEC's permittivity and $\sigma_{THz}$, using the experimental values obtained from the THz-TDS measurements on the bare p(g2T-TT)−see Supplementary Fig. 4. Figure 2d, e display the results of the simulations, which show that while the extent of the modulation appears to be similar to the experimental one, the simulation data are however shifted towards more negative gating voltages, and they exhibit a steeper negative slope. This discrepancy is to be expected, and it is due to the device architecture used for the characterization of the bare polymer, which is essentially an OECT characterized by a large channel length (i.e. 2 mm)[37]. Indeed, stronger gate biases are needed to achieve a homogeneous accumulation of carriers in this device configuration since charge injection occurs only at the two widely separated bottom electrodes, while in the metadevice this is not the case, as the inverse metasurface acts as a full back electrode for the entire OMIEC layer. Moreover, the increased thickness of the OMIEC layer in the OECT with respect to the metadevice (i.e. 850 nm versus 60 nm) also contributes to the shift, as well as to the increased steepness, of the simulation data (see ref. 37 and Supplementary Fig. 2).

Overall, these results confirm that THz conductivity modulations of p(g2T-TT), albeit lower than what is typically measured in dc for this class of OMIECs, are sufficient for achieving large MD in the metadevices.

## Direct-written fabrication of metadevices

Finally, to showcase the potential of this materials approach for rapid and versatile manufacturing, we implemented the same metadevice architecture solely by means of direct-writing techniques. This is the first implementation of an electrically-tuned THz modulator that is realized exclusively with this set of techniques. The proposed process flow, while combining sequential methods, can in principle be scaled to mass-manufacturing, as it combines two highly parallelizable technologies based on commercial tools compatible with large-area fabrication−i.e. inkjet printing[26] and laser ablation[27]. In particular, the reliability of printing processes for electronics manufacturing is here strengthened through the synergy with the laser ablation step for the

definition of critical features, which offers an inherently self-aligned approach. Moreover, our approach is potentially compatible with alternative printing or coating techniques, such as screen-printing, which are mature technologies for the realization of electronic components[38].

A commercial silver nanoparticle suspension was inkjet-printed and subsequently sintered on the substrate of choice−either fused silica or plastics−thus forming two coplanar electrodes of approximately ∼200 nm in thickness. The complementary metasurface was then obtained via femtosecond laser ablation of the inner square electrode, hence replicating the inverse design detailed in Supplementary Fig. 1. An OMIEC layer of ∼60 nm was then inkjet printed on both the metasurface and the lateral gate electrode, and subsequently covered with the gating iongel electrolyte. As in the case of the microfabricated metadevice, the addition of these two final layers decreases both the resonance frequency and the base transmission of the metasurface because of their complex permittivities. Interestingly, the use of printing and direct-writing techniques does not negatively affect the quality factor, nor the modulation, of the inverse metasurface. On the contrary, in comparing the base transmission of bare metasurfaces fabricated with either photolithography or laser writing, we find that the latter exhibits an enhanced quality factor of the resonance−see Supplementary Fig. 5. This result is possibly due to the conductivity of the silver film being higher than the gold one, and confirms the overall quality of the laser machining process (Fig. 3b).

The THz-TDS measurements performed on the fully direct-written samples confirm the results previously obtained with the metadevice realised using conventional microfabrication. The MD and Δφ are again assessed at ∼65% and 70° on the rigid substrate, and at ∼57% and 65° on the flexible one−see Supplementary Fig. 6.

In summary, our results introduce organic electronic materials as an effective tool for the realisation of reconfigurable THz optics. Indeed, the extreme hole density modulations occurring in a model organic mixed conductor are shown to elicit substantial charge screening effects in the 0.5–1.5 THz range, despite OMIECs having relatively limited charge mobilities with respect to inorganic and 2D materials. These results constitute a compelling opportunity for the development of THz optics with unique form-factors−such as flexible, ultrathin, conformable, or bioinspired devices−thanks to the ease of deposition and patterning that is granted by solution-phase and direct-writing methods. The environmental footprint of the proposed approach will further benefit from the use of environmentally friendly solvents for processing the functional layers, where aqueous inks represent a promising opportunity[39], or from the implementation of liquid extraction methods for the recycling of polymeric materials[16]. At the same time, compatibility of the present process with flexible low-budget substrates, which represent by far the largest volume in a device, will favour the use of recycling schemes, as well as biodegradable and compostable solutions.

From an operational point of view, a more in depth understanding of THz charge transport in these materials is needed to fully harness this tuning strategy. Some of the open questions emerging from this work relate, for example, to the microscopic interpretation of the Drude-Smith localization parameter in organic semiconductors, as well as to the nature of localized carriers at different gating voltages, and to what extent these effectively contribute to screening effects. A possible route to further investigate this matter would involve a combination of electrical, structural, and THz characterization techniques performed on a larger pool of OMIECs with different microstructural characteristics, possibly paired with a detailed microscale modelling of the light-matter interactions taking place at this frequency range. Moreover, dedicated efforts should be devoted to the characterization and optimization of the modulation speed of these devices, that is the time needed for the devices to switch between different transmission levels, in order to better establish their potential

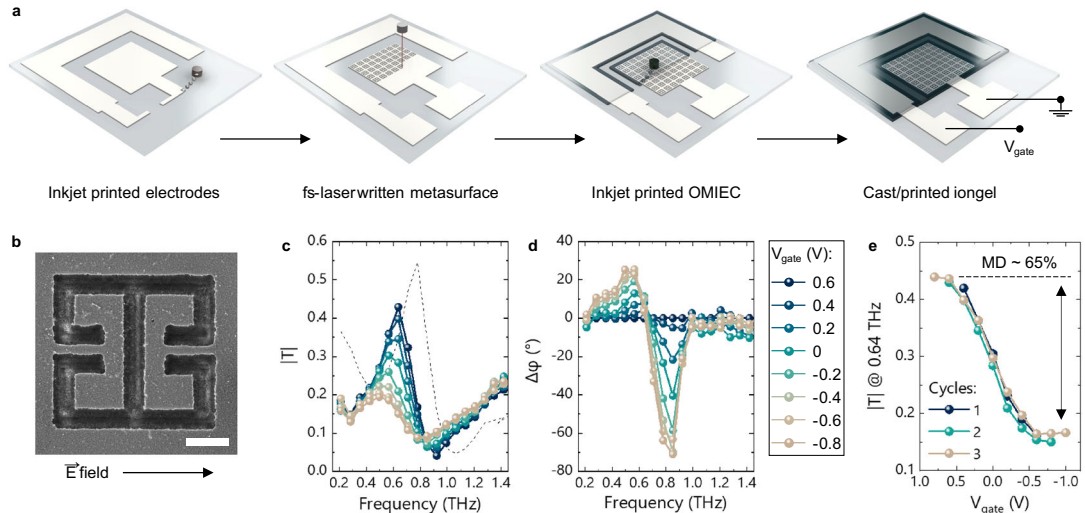

**Fig. 3 | Fully direct-written metadevices. a** Schematic representation of the fabrication steps employed for the realization of the OMIEC-based tuneable metasurface. **b** Image of an individual *inverse* electrical SRR obtained by fs-laser ablation on an inkjet printed silver electrode: the arrow displays the orientation of the in-plane electric field polarization (scale bar corresponds to 10 μm). Amplitude

and phase modulation of the complex THz transmission, where the magnitude (**c**) is calculated with respect to air, while the phase (**d**) uses the 0.6 V spectrum as reference. The dashed line in **c** corresponds to the transmission of the metasurface without OMIEC and iongel. **e** Voltage dependence of the transmission at -0.64 THz for three consecutive measurement cycles.

impact in key application scenarios such as wireless communications. A more complete understanding of these fundamental matters would be instrumental for advancing and optimizing this novel tuning strategy, hence opening a new set of opportunities and tools for scientists and engineers designing the new generations of THz technologies.

## Methods

### Materials
Fused silica substrates (thickness: 1 mm) were purchased from FOCtek Photonics. Polyimide substrates (≈125-μm thick) were purchased from Sigma-Aldrich, and used without cleaning. Ionic liquid 1-ethyl-3methylimidazolium bis(trifluoromethylsulfonyl)imide (EMIM:TFSI) was purchased from Sigma-Aldrich. Polymeric insulator poly(styrene-b-methyl methacrylate-b-styrene) powder with an average Mn: PS(4500)-b-PMMA(19000)-b-PS(4500) was purchased from Polymer Source. The silver nanoparticle dispersion (30−35 wt% in triethylene glycol monoethyl ether) was purchased from Sigma-Aldrich.

### Fabrication
The metadevices realised with conventional microfabrication techniques consisted of a photolithographically patterned Cr/Au (5/100 nm) layer, on which the OMIEC−i.e. p(g2T-TT) in chloroform, 8 mg/ml−was spin-cast at 1000 rpm for 1 min, briefly annealed at 80 °C to dry the solvent, and then patterned with a laboratory swab.

The direct-written metadevices were fabricated by inkjet printing a 200 nm layer of silver onto both fused silica and polyimide substrates, followed by a sintering step at 180 °C for 1 h. The inverse metasurface design was then obtained via femto-second laser ablation.

The iongel solution was obtained by mixing 900 μL of n-Butyl Acetate with 90 μL of EMIM:TFSI liquid salt and 10 mg PS-PMMA-PS. The ion gel electrolyte was then deposited over the semiconductor and annealed at 120 °C for 1 h.

The laser micromachining system used to realize the inverse metasurface consisted of an amplified Yb:KGW femtosecond laser system (Light Conversion, Pharos) with pulses of 240 fs duration, amplified to the fundamental wavelength (1030 nm), with a repetition frequency of up to 1 MHz and a pulse energy of up to 0.2 mJ. A wavelength of 515 nm (second harmonic) was used to perform the ablation. The laser light was statically focused on the substrate surface through a microscope objective (20X-0-40 NA, M Plan Apo SL20X, Mitutoyo) and

the 2D structure was obtained by moving the sample. To translate the sample with respect to the desired path and with the correct pulse deposition energy density, computer-controlled 3-axis translation stages (ABL-1000, Aerotech) with a maximum resolution of a few tens of nm over a wide range were used, interfaced by CAD-based software (ScaBase, Altechna) with an integrated acousto-optic modulator. An average power (pulse energy) of 40 mW (80 nJ), a repetition frequency of 500 kHz, a pulse density per millimetre of 2500 and a scanning speed of 5 mm/s were used.

The OMIEC was then inkjet-printed onto the metasurface and gate electrode using a commercial machine (Dimatix DMP 2831) with an ink composition of p(g2T-TT) in chlorobenzene:dichlorobenzene (3:1) at 1.5 mg/ml. After an annealing period at 120 °C for 1 hour in nitrogen atmosphere, the iongel was deposited and subsequently annealed.

The OECT used for the electrical and THz-TDS measurements on the bare polymer was fabricated with the same protocol used for the direct-written devices. The *W/L* for this device is 3 mm/2 mm, as to grant sufficient channel area to avoid scattering of the THz beam with the source and drain metal electrodes.

### Electrical characterisation
Measurements of the OECT characteristic curves were performed in inert atmosphere (nitrogen filled glovebox, $O_2$, and $H_2O$ levels below 1 ppm) by means of an Agilent B1500A Semiconductor Parameter Analyzer. The lateral drain-source voltage was set at −0.2 V with a scan speed of 2 mV/s and an initial hold time of 60 s.

### Terahertz spectroscopy
The THz time-domain spectroscopy setup is driven by a Ti:Sa amplified laser system, delivering 50-fs pulses with a repetition rate of 1 kHz centred at λ ≈ 790 nm. A pulse energy of approximately 1 mJ is delivered to the THz setup and split into two lines devoted to THz generation and THz detection. For this experiment, THz pulses are generated by optical rectification in a ZnTe crystal, 1 mm-thick and <110 >-oriented. This gives a 1 kV/cm electric field probe with a bandwidth between 0.4 THz and 1.5 THz, which is directed to the sample by off-axis parabolic mirrors, obtaining a probe spot-size at the sample of ~0.7 mm. The transmitted THz pulse is then reconstructed in the time domain by electro-optic sampling (EOS), exploiting the interaction of the THz electric field with the gate pulse in a 1 mm-thick and <110 >-

oriented ZnTe detector crystal. The sample is at room temperature and the THz pulses propagate in a chamber purged by dry-nitrogen, to avoid THz absorption by water vapour. We obtained the THz complex transmission by acquiring both a time-domain trace of the THz wave interacting with the sample, and a reference one. For the measure on the metadevices, the reference was the complex spectrum recorded without the metadevice, while in the case of the bare OMIEC measurements reference was the complex spectrum transmitted by the substrate covered with the iongel. To determine the conductivity of the sample, we performed a numerical optimization technique minimizing the difference between the experimentally extracted transmission and a modelled transmission accounting for the light travelling inside our sample. From this procedure, we can obtain the sample's complex refractive index from which we can extract the THz conductivity.

### Simulations

The numerical calculations were performed using the time domain solver of HFSS Ansys with periodic boundary conditions.

## Data availability

All data used in this study, when not already presented in the main text or in the Supplementary Information, are available from the corresponding authors upon request.

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

## Author contributions

G.E.B. conceived the idea, designed and simulated the metadevice architecture. C.K. synthesised the organic mixed conductor. F.M. and G.E.B. fabricated the photolithographic metadevices. C.B. developed the OMIEC-based ink, optimized the inkjet printing of the metallic and polymer layers, and fabricated the direct-written devices. M.B. optimized and performed the femto-second laser machining process, and took the SEM image of the direct-written metasurface. C.B. and G.E.B. performed the electrical characterization. F.G., L.G., G.E.B. and C.B. performed the terahertz measurements. F.G. and E.C. performed the analysis and fit of the terahertz conductivity data. G.E.B., E.C., M.C., C.V., I.M. designed the experiments, analysed the data, and supervised the activities. All authors have contributed to writing the manuscript. C.B. and F.G. contributed equally to this work.

## Competing interests

A patent application has been filed by Tufts University listing G.E.B. and Fiorenzo Omenetto as inventors of a metadevice technology based on organic mixed-conductors (US Patent App. 17/449,935). The remaining authors declare no competing interests.
