## [Transparent Peer Review file · Nature Communications]

Tuning direct-written terahertz metadevices with organic mixed ion-electron conductors

Corresponding Author: Professor Giorgio Ernesto Bonacchini

Version 0:

Reviewer comments:

Reviewer #1

(Remarks to the Author)

This work focuses on exploring and using a type of organic material in high-frequency electronic and light-based devices, particularly in the terahertz (THz) range. Over the past ten years, these materials have been successfully used in several cutting-edge technologies, including medical devices, brain-inspired systems, light-based devices, energy collection, and printed electronics.

The paper is well written and thorough, and it is a very well carried out exploration that could indeed lead to new directions in organic electronics emphasizing their use in high-frequency areas like microwave and terahertz ranges, which has traditionally been limited.

The work is thorough and of high enough impact that it merits publication in Nature Communications. There are no particular comments on the fabrication and measurements that are relevant/substantial enough to point out. The work is very elegant and clever and it is rigorously carried out.

To make the paper more relevant to the broader audience of Nature Communication readers, it would be helpful for the authors to better frame and clarify some of the key issues they've outlined. I believe addressing this would enhance the paper's impact and understanding.

The work emphasizes the reduced environmental footprint, and in technological contexts such as these, framing results as "green" requires a balanced and realistic assessment of the entire lifecycle of these materials—from production to disposal—ensuring that they remain environmentally friendly. This could be a challenge, especially if they were adopted on a larger scale. Reframing the approach to identify practically the challenges in this context would make the paper stronger.

From a technical perspective, along these lines, scaling up the production of these new metasurfaces while maintaining their flexibility, light weight, and unique mechanical properties could present challenges, particularly in ensuring consistency and reliability at a mass scale. Again, a phrase of two analyzing the reliability issues and how to address them in the future, combined with the considerations above would add more strength to the work.

What issues are expected in integrating these organic materials with existing high-frequency devices and systems?

Could the authors expand on the suitability of the the Drude-Smith model to simulate the devices presented here? Literature has shown its use in disordered media, as the authors point out, yet fully capturing the dynamics of the complex and diverse scattering mechanisms present in semiconductive polymers (trapping, hopping, etc) and potential polymer degradation, conformational changes may lead to challenges.

Reviewer #2

(Remarks to the Author)

The authors introduce a new device based on OMIECs. The authors correctly state OMIECs have been demonstrated in a lot of applications, but this is the first time those materials have been used for THz applications. The importance being that this frequency can be used for sensing and telecom applications. The authors use reasonably accessible deposition technologies, which are also compatible with low temperature and therefore low-cost substrates.

Organic electronic materials are generally dismissed for, and therefore not commonly used for, high frequency applications. It is important to address, communicate and explain how other researchers can develop the area of organic electronics in high frequency applications. I really like the work; it is novel, and the manuscript is structured well. I think the work is suitable for Nature Communications and has the necessary impact and experimental sounding to introduce a new idea, but I'm uncertain about the citations or the work that will come from this, and that the manuscript will be overlooked, because of the current explanation. In this paper, the authors have the potential to start a new field by merging high Hz with organics – but as the paper stands, it is not general or accessible, which is needed to continue a new field or application space.

I recommend publication if the authors can be clearer, in recognition of their broad audience and the impact their work could have. Some specific points are as follows:

1. "We employ the polymer p(g2T-TT) – the prototype compound for this class of OMIECs– which we deposit for the first time by inkjet printing using a commercial printer." The authors might consider re-phrasing this sentence. It's not the first time a mixed conductor has been ink jet printed, and considering their overall application, it's not relevant to state it is the first time p(g2T-TT) has been ink jet printed.
2. There are no preconceptions as to why organics aren't used for high frequency devices, as the authors suggest (e.g., "the preconception that organic materials have no utility in high-frequency applications). Charge carriers move through organic materials slowly, and therefore do not respond on a time that is equivalent to a high frequency. The following language simplifies the limitation of organics for high frequency applications. It is shallow and confusing and needs to be rephrased: "A possible explanation of this gap in both scientific literature and technology is likely due to the relatively low charge carrier mobilities and poor charge injection of organic semiconductors with respect to other families of semiconductors . This limitation has possibly originated the preconception that organic materials have no utility in high-frequency applications."
3. On the same line as the previous comment: the literature suggests OMIECs have much slower charge transport than OSCs. How and why is a mixed conductor needed (or better) than an OSC?
4. Figures are most important, yet here, not clear. Figure 2a – there is a green cylinder coming out of a planar structure, and has no label and therefore no meaning? If you read the text – it could even be a third electrode.
5. Can we explain the difference between the "metastructure", that the authors specify is not Figure 2, to the later "metadevice structure" they talk about later? Can we be clearer on what is the difference between metastructure and metadevice structure and switching between these terms? Can the authors build on these device components a little better? What matters more – metadevice or metastructure?
6. The three electrode device – and why is it removed later? Was it a transistor? Why do the authors talk about a gate voltage? Why don't the diagrams match up to that?

Reviewer #3

(Remarks to the Author)
As attached in the file.

Version 1:

Reviewer comments:

Reviewer #1

(Remarks to the Author)
I am fully satisfied with the answers that the authors provided to the various referee comments and think the paper is ready for publication.

Reviewer #2

(Remarks to the Author)
The authors have provided detailed and interesting responses to the reviewer questions. However, a lot of these great answers then don't translate into the main text. It would help the reader a lot to have key points from the information in the response, in the manuscript.

Also, Figure 2a still needs to be labelled for clarity.

Reviewer #3

(Remarks to the Author)
Thank you for your responses.

Response to the reviewers:

We thank the reviewers for assessing our work and for their useful and detailed comments. Here below we address one-by-one all the raised points and, where applicable, we indicate how the manuscript was amended.

Reviewer #1:

This work focuses on exploring and using a type of organic material in high-frequency electronic and light-based devices, particularly in the terahertz (THz) range. [...] The work is thorough and of high enough impact that it merits publication in Nature Communications. There are no particular comments on the fabrication and measurements that are relevant/substantial enough to point out. The work is very elegant and clever and it is rigorously carried out.

We thank the reviewer for the positive comments on the quality of our work and of the manuscript.

To make the paper more relevant to the broader audience of Nature Communication readers, it would be helpful for the authors to better frame and clarify some of the key issues they've outlined. I believe addressing this would enhance the paper's impact and understanding.

We thank the reviewer for the excellent suggestions on how to better frame our work to the wide and diverse readership of Nature Communications. Both in this document and in the amended manuscript, we have endeavored to clarify the raised points – see below.

The work emphasizes the reduced environmental footprint, and in technological contexts such as these, framing results as “green” requires a balanced and realistic assessment of the entire lifecycle of these materials—from production to disposal— ensuring that they remain environmentally friendly. This could be a challenge, especially if they were adopted on a larger scale. Reframing the approach to identify practically the challenges in this context would make the paper stronger.

While a quantitative life-cycle analysis (LCA) is clearly out of the scope of our study, we agree with the reviewer that our claims of sustainability would benefit of a more in-depth discussion of the actual advantages and challenges of the proposed deposition methods. We have hence added the following sections (in red) to the amended manuscript:

- **Introduction** (page 3):

Thanks to this approach, we are able to fabricate THz metadevices where both the metasurface and the active materials are processed solely by cost-effective and mass-scalable direct-writing techniques, namely inkjet printing and femtosecond laser writing, which are also compatible with large-area and plastic substrates²⁵. These methods offer a maskless process flow that ensures an efficient use of materials, minimizing waste. Combined with the low temperature processing allowed by OMIECs, this fabrication scheme relies on a reduced energy consumption and relatively low capital expenditure, thus contributing to its sustainability^{26,27}. At the same time, the absence of critical raw materials is a further advantage²⁸.

- **Conclusions** (page 10):

These results constitute a compelling opportunity for the development of THz optics with unique form-factors – such as flexible, ultrathin, conformable, or bioinspired devices – thanks to the ease of deposition and patterning that is granted by solution-phase and direct-writing methods. The environmental footprint of the proposed approach will further benefit from the use of environmentally friendly solvents for processing the functional layers, where aqueous inks represent a promising opportunity³⁹, or from the implementation of liquid extraction methods for the recycling of polymeric materials¹⁶. At the same time, compatibility of the present process with flexible low-budget substrates, which represent by far the largest volume in a device, will favor the use of recycling schemes, as well as biodegradable and compostable solutions.

From a technical perspective, along these lines, scaling up the production of these new metasurfaces while maintaining their flexibility, light weight, and unique mechanical properties could present challenges, particularly in ensuring consistency and reliability at a mass scale. Again, a phrase of two analyzing the reliability issues and how to address them in the future, combined with the considerations above would add more strength to the work.

As with the previous comment, we find that the objective assessment of future industrial/large-scale production schemes and their reliability is a task that, given its complexity, transcends the scope of our investigation. Nonetheless, we fully agree with the reviewer that readers unfamiliar with direct-writing technologies would benefit from a clearer outline of the potential of these techniques for mass-scale production. We have thus added the following sentences (in red) to the section “**Direct-written fabrication of metadevices**” at page 8:

The proposed process flow, while combining sequential methods, can in principle be scaled to mass-manufacturing, as it combines two highly parallelizable technologies based on commercial tools compatible with large-area fabrication – i.e. inkjet printing²⁶ and laser ablation²⁷. In particular, the reliability of printing processes for electronics manufacturing is here strengthened through the synergy with the laser ablation step for the definition of critical features, which offers an inherently self-aligned approach. Moreover, our approach is potentially compatible with alternative printing or coating techniques, such as screen-printing, which are mature technologies for the realization of electronic components³⁸.

What issues are expected in integrating these organic materials with existing high-frequency devices and systems?

Given the wide and diverse set of microwave/terahertz technologies brought forward in the past decade, the answer to this question very much depends on the specific application scenario as well as the exact frequency range and functionality that is envisioned¹. Indeed, depending on whether we discuss about telecommunication, sensing, imaging, computing or cloaking technologies, the integration of organic metadevices in existing (or emerging) microwave/terahertz systems may face different types of hurdles, which can be either operational in nature, or related to fabrication constraints.

Let us consider as an example the realm of high-speed wireless communication, where metadevices are displaying a remarkable potential for the development of compact antennas and highly efficient beam steering platforms². In this context, among the primary factors that need to be investigated and optimized for the successful integration of OMIEC-based metadevices/optics into such systems is the *modulation speed*, that is the time needed for the devices to switch between different transmission levels. From the extensive literature on OECTs, the response times of OMIEC-based electronic devices are typically limited to few tens of microseconds, which are too low for applications that require modulation frequencies above few tens of kilohertz³. Nonetheless, recent advances have pushed this limit to sub- μ s switching times through innovative materials and device configurations⁴, although it is not yet clear whether these advancements could be implemented in metadvice architectures as well. As discussed in response to comment 1 from reviewer #3, the response time of our devices was not studied nor optimized due to the long acquisition times of our current THz setup, although we do plan to investigate this aspect in future works.

From a fabrication standpoint, another factor that might represent a challenge in certain applications is the compatibility of organic electronic materials with back-end-of-the-line fabrication processes. Indeed, since established microwave/terahertz technologies rely primarily on inorganic semiconductors and materials fabricated using conventional fab techniques, the integration of polymers into existing device stacks might require the development of materials that withstand high temperatures during fabrication. While we believe it is quite early to address this aspect, this problem is starting to raise attention in other emerging – yet more established – applications of OMIEC materials, see for example the interesting review article by A. Gumyusenge et al. on materials strategies for organic neuromorphic devices⁵.

Overall, while we perfectly agree with the reviewer that this question could be of great interest for the reader, given the complexity of the topic we believe that an appropriate answer requires an extended analysis, for

example in the form of a minireview. We hence limited our intervention on the manuscript to the following addition to the **Conclusions** of the manuscript (in red):

From an operational point of view, a more in depth understanding of THz charge transport in these materials is needed to fully harness this tuning strategy. Some of the open questions emerging from this work relate, for example, to [...]. Moreover, dedicated efforts should be devoted to the characterization and optimization of the modulation speed of these devices, that is the time needed for the devices to switch between different transmission levels, in order to better establish their potential impact in key application scenarios such as wireless communications. A more complete understanding of these fundamental matters would be instrumental for advancing and optimizing this novel tuning strategy, hence opening a new set of opportunities and tools for scientists and engineers designing the new generations of THz technologies.

Could the authors expand on the suitability of the Drude-Smith model to simulate the devices presented here? Literature has shown its use in disordered media, as the authors point out, yet fully capturing the dynamics of the complex and diverse scattering mechanisms present in semiconductive polymers (trapping, hopping, etc) and potential polymer degradation, conformational changes may lead to challenges.

We thank the reviewer for the insightful question. When carrier localization occurs, the phenomenological Drude-Smith model has been shown to accurately reproduce the THz conductivity of various systems with structural disorder⁶. While a more detailed model, such as an effective medium theory combined with chain-to-chain hopping, could offer additional insights into intra- and inter-polymer conductivity, it would also introduce more free parameters without significantly improving the fit beyond what the Drude-Smith model already provides. Moreover, effective medium theory is typically applicable to systems with low filling factors (dispersed media) and to particles with simple shapes such as spherical, cylindrical, or flat disks, which do not match the characteristics of organic semiconductors. A comparison between the Drude-Smith model and effective medium theory plus hopping is discussed in Ref. ⁶, particularly in Fig. 14.

For conductive polymers with conductivity at the kS/m level, the localized-modified Drude theory was demonstrated to be effective in modeling the THz conductivity and returns reliable localization parameters⁷. In this case, there are no specific advantages with respect of Drude-Smith as the localized-modified Drude model has an extra parameter to consider localization that, for conductive polymers, is expected to be of the order of unity.

However, the primary focus of this paper is to demonstrate a proof of principle for this type of THz modulating device, which has not been applied in this spectral region before. For this reason, we chose the phenomenological Drude-Smith model to gain a general understanding of the observed phenomena. Notably, a similar approach was already used in Refs. ^{8,9}.

Finally, as mentioned in the conclusions, we believe that a thorough THz investigation of the short-range charge transport dynamics in OMIECs requires a combination of structural, electrical and spectroscopic techniques, and they are thus worthy of dedicated efforts and resources. For example, in our future studies we aim at probing the THz conductivity of OMIEC films with different microstructure and molecular weight^{11,12}, as well as at investigating the presence (and the causes) of short-range transport anisotropies through the alignment of the polymer chains¹³. We believe that these two methodologies, through the appropriate correlation with conventional electrical measurements, will provide an even deeper understanding of the charge transport properties of this class of materials while also further elucidating the actual suitability of the available transport models, such as Drude-Smith, for organics.

Reviewer #2:

The authors introduce a new device based on OMIECs. [...] I really like the work; it is novel, and the manuscript is structured well. I think the work is suitable for Nature Communications and has the necessary impact and experimental sounding to introduce a new idea, but I'm uncertain about the

citations or the work that will come from this, and that the manuscript will be overlooked, because of the current explanation. In this paper, the authors have the potential to start a new field by merging high Hz with organics – but as the paper stands, it is not general or accessible, which is needed to continue a new field or application space.

We thank the reviewer for appreciating the potential of our work. We also acknowledge their reasonable concerns on the accessibility of our study to the wider audience, similarly expressed by Reviewer #1, and we have thus endeavored to clarify some general aspects discussed in our manuscript.

I recommend publication if the authors can be clearer, in recognition of their broad audience and the impact their work could have. Some specific points are as follows:

1. “We employ the polymer p(g2T-TT) – the prototype compound for this class of OMIECs– which we deposit for the first time by inkjet printing using a commercial printer.” The authors might consider re-phrasing this sentence. It’s not the first time a mixed conductor has been ink jet printed, and considering their overall application, it’s not relevant to state it is the first time p(g2T-TT) has been ink jet printed.

We agree with the reviewer that p(g2T-TT) is not the first OMIEC to be inkjet-printed. Indeed, a variety of printable formulations of PEDOT:PSS – the archetypical OMIEC used in organic (bio)electronics – are commercially available and used in both academic and industrial settings. However, for what concerns the specific type of OMIEC discussed in this work (i.e. conjugated polymers with glycolated sidechains), we believe that our work is the first to report material deposition using this particular technique. Indeed, the excerpt quoted by the reviewer in their comment immediately follows from another sentence in the manuscript, where this particular subclass of OMIECs is singled out among other materials – see below:

We take advantage of an emerging class of OMIECs characterized by conjugated polymer backbones that exhibit high hole (or electron) mobilities, while also displaying liquid-like ionic conduction thanks to their hydrophilic glycolated sidechains²³.

Nonetheless, to avoid possible misunderstandings, we have decided to remove the locution “for the first time” from the revised manuscript.

2. There are no preconceptions as to why organics aren’t used for high frequency devices, as the authors suggest (e.g., “the preconception that organic materials have no utility in high-frequency applications). Charge carriers move through organic materials slowly, and therefore do not respond on a time that is equivalent to a high frequency. The following language simplifies the limitation of organics for high frequency applications. It is shallow and confusing and needs to be rephrased: “A possible explanation of this gap in both scientific literature and technology is likely due to the relatively low charge carrier mobilities and poor charge injection of organic semiconductors with respect to other families of semiconductors . This limitation has possibly originated the preconception that organic materials have no utility in high-frequency applications.”

We understand and accept the critique concerning our use of the word *preconception*, which is not rigorous and may indeed constitute an overly simplistic and subjective motivation for the scarce use of organic electronic materials in high frequency applications, particularly to a non-specialized readership. We thank the reviewer for pointing this out.

We have thus modified the whole final portion of this section as follows (red text below):

A possible explanation of this gap in both scientific literature and technology is likely due to the relatively low charge carrier mobilities and poor charge injection of organic semiconductors with respect to other families of semiconductors, which result in relatively low device switching speeds¹⁸. This limitation is likely the reason for the scarce use of organic materials in high-frequency applications^{19,20}.

3. On the same line as the previous comment: the literature suggests OMIECs have much slower charge transport than OSCs. How and why is a mixed conductor needed (or better) than and OSC?

As noted by the reviewer, OMIECs are typically characterized by inferior electron/hole mobilities with respect to OSC. Nonetheless, transistor based on mixed conducting polymers – i.e. OECTs – exhibit much higher transconductances and larger channel conductivities (in accumulation) with respect to other organic transistors technologies, despite the relatively low charge mobilities. Indeed, since the transconductance g_m is linearly proportional to both the specific capacitance and the mobility, the relatively poor charge transport properties of this class of materials are compensated by the massive capacitive coupling established between the electrolyte and the OMIEC, thanks to the mixed transport characteristics of the former.

In a similar way, the attenuation of the transmission displayed by our devices is determined by the changes in the imaginary part of the OMIEC complex permittivity. This parameter is linearly proportional to the *conductivity* of the polymer, hence encompassing both the *carrier mobility* AND the charge *carrier concentration*. It is hence likely that the large charge density modulations obtained across the OMIECs volume balance the relatively low mobility of these materials with respect to OSCs, where the accumulated charge is restricted to a nanometric surface layer at the OSC-electrolyte/insulator interface.

Moreover, we wish to note that the mobility values reported for OSCs and OMIECs in the current literature are measured in DC, and thus relate to the long-range transport of hole (or electron) charges, often embedding the effect of suboptimal charge injection from the contacts¹⁴. In our THz measurements, the motion of the mobile charges is probed with oscillating lateral fields lasting few picoseconds, and thus shed light on the short-range transport characteristics. Given the big difference between these two regimes, the *a priori* assumption that OMIECs have a lower terahertz mobility is not necessarily correct, particularly since we know so little of the terahertz properties of organic electronic materials in general. As discussed in the manuscript, it is indeed our hope that this work will provide a basis for future investigations in this direction, with the goal of clarifying these fundamental aspects which have important technological implications.

4. Figures are most important, yet here, not clear. Figure 2a – there is a green cylinder coming out of a planar structure, and has no label and therefore no meaning? If you read the text – it could even be a third electrode.

We thank the reviewer for this note. The green cylinder in Figure 2a and Supplementary Figure 4 graphically represents THz beam, thus indicating the area of the device where the THz properties are measured. We have clarified this by adding a label to the panel/inset in both Figures:

5. Can we explain the difference between the “metastructure”, that the authors specify is not Figure 2, to the later “metadevice structure” they talk about later? Can we be clearer on what is the difference is between metastructure and metadevice structure and switching between these terms? Can the authors build on these device components a little better? What matters more – metadevice or metastructure?

The appellation *metastructure*¹⁵ refers to any object based on the metamaterial paradigm, that is the ordered assembly of meta-units. This term thus encompasses *metamaterials*, *metasurfaces*, and other somewhat exotic device concepts (e.g. *metagratings*¹⁶). In our manuscript, it is used as a synonym to the word *metasurface* to avoid repetitions.

On the other hand, the term *metadevice* indicates active or passive reconfigurable/tuneable metastructures¹⁷. With the locution *metadevice structure* we thus refer to the architecture and geometry of our device intended as the specific stack of its constituting layers: substrate (plastic or glass), metasurface with lateral gate electrode (silver or gold), OMIEC (printed or coated), and electrolyte.

We acknowledge that the abundance of somewhat similar terms may be confusing to the reader, and for this reason in the amended manuscript we have substituted all references to *metadevice structure* with *metadevice architecture*, as well as the term *metastructure* with the word *metasurface*.

6. *The three electrode device – and why is it removed later? Was it a transistor? Why do the authors talk about a gate voltage? Why don't the diagrams match up to that?*

The three-electrode device shown in Figure 2a is indeed a transistor, as it is composed of two metal electrodes bridged by the OMIEC – i.e. the source and drain contacts – a coplanar gate electrode and an electrolyte as gating medium. In the context of this work, this device was used for two purposes:

1. The extraction of the DC conductivities at different gate voltages, as reported in Supplementary Fig. 3. In this instance, the device was operated as a conventional transistor by applying a lateral voltage V_{DS} along with the gate polarization. The σ_{DC} values in the plot hence refer to the OMIEC conductivity in the OEET channel, and they were calculated from the drain-source currents I_{DS} , knowing both V_{DS} and the geometry of the channel. As discussed in the text, these results are used to explain the difference between the simulated and the experimental modulation curves – see Figure 2e.
2. The THz-TDS characterization of the bare OMIEC, at different gating voltages, without the underlying metasurface. Since our terahertz setup presently allows for transmission measurements only, it was necessary to implement a system of electrodes that would allow the charging of the polymer without obstructing the THz beam – hence the relatively long channel ($W/L = 3 \text{ mm} / 2 \text{ mm}$), since the spot size of the THz beam is approximately 0.7 mm in diameter. By shorting and grounding the drain and source contacts, the device effectively operates as a two-electrode system where the OMIEC is charged through both contacts. This configuration allows us to probe the THz conductivity of the mobile charges within the transistor channel, even though no lateral DC current is present since $V_{DS} = 0 \text{ V}$.

In order to better clarify these points to the reader, we have amended the manuscript in the following sections (see in red the added text):

- **Evaluation of terahertz charge transport dynamics in the polymer** (page 5):
With this device architecture, the three terminals can either be connected and biased as the gate, drain, and source electrodes of a conventional organic electrochemical transistor (OEET)³⁰, or operated as a two-electrode system by grounding both the drain and source contacts – i.e. $V_{drain} = V_{source} = 0 \text{ V}$. The first configuration allows the extraction of the steady-state electrical conductivity σ_{DC} of the OMIEC from the characteristic transfer curve of the OEET, shown in Supplementary Fig. 2, while through the latter we perform *in situ* spectroscopic measurements of the THz conductivity σ_{THz} at different electrostatic doping levels, thus capacitively charging the OMIEC without obstructing the THz beam.
- **Methods, Fabrication** (pages 10-11):
The OEET used for the electrical and THz-TDS measurements on the bare polymer was fabricated with the same protocol used for the direct-written devices. The W/L for this device is 3 mm / 2 mm, as to grant sufficient channel area to avoid scattering of the THz beam with the source and drain metal electrodes.

For what concerns the plot and diagrams in the manuscript, for the sake of simplicity and consistency, we have decided to maintain the terminology *gate voltage* and V_{gate} not only when referring to the OEET (whether in the two- or in the three-electrode configuration), but also for the metadevices. This choice is further motivated by considering that in the equivalent circuit model for the reconfigurable meta-unit – i.e. the inverse split-ring resonator shown in Supplementary Fig. 1 – the gate-controlled tunable resistance can be envisioned as a transistor – see Ref. ¹⁸.

Note: the positions of Supplementary Fig. 2 and Supplementary Fig. 3 have been inverted with respect to the original manuscript according to the new referencing order.

Reviewer #3:

The manuscript titled "Tuning direct-written terahertz metadevices with organic mixed ion-electron conductors" presents the usage of organic materials in combination with metasurfaces on both rigid and flexible substrates. [...] As the paper demonstrates potential, I recommend acceptance with revisions. I have some suggestions and questions to the authors to improve the paper.

We thank the reviewer for the positive feedback and for the useful suggestions.

1. What is the response time of the tuneable metasurface?

Given the long acquisition time for the TDS-THz signal in our setup (several minutes), we are not presently able to quantify the actual response time of our optical devices, which is faster. Indeed, the plot below reports the charging current versus time of our device upon a V_{gate} step of -200 mV, which can be used to estimate the switching time. By fitting this data with a double exponential, we find that the fast decay occurs with a timescale of 0.63 s, while the slower one has a time constant of approximately 6.42 s.

Note that in line with what is achieved with transistors, the switching speed can be optimized by adjusting the thickness of the OMIEC layer, achieving much faster time constants¹⁹. However, our device was not optimized for this figure of merit in this work.

2. Can the authors further comment on the advantages of using the organic mixed ion-electron conductors as opposed to other media for tuning metasurfaces such as graphene?

As discussed in response to comment 3 from Reviewer #2, the unique characteristic of OMIECs is the possibility to modulate the charge carrier density (i) extensively (up to 20 orders of magnitude) and (ii) across the entire volume of the material. On the other hand, in other semiconductors (e.g. conventional organics, 2D materials and carbon nanotubes) the charge is accumulated only as a surface layer at the interface with the gating medium, hence providing an inferior modulation of the overall conductivity of the active layer in the device. In addition, graphene FETs typically display low ON/OFF currents (i.e. low charge carrier modulation) due to the semimetallic nature of this material²⁰. As an example, see Ref. ²¹ for a graphene FET with similar architecture to our three-terminal device, showing a maximum ON/OFF ratio of 400 against the four orders of magnitude obtained with pg(2T-TT).

Moreover, while there is ample availability of printable graphene formulations, these materials do not yet exhibit the same desirable charge transport properties of the graphene used in metadevices²², whose realization relies on complex chemical vapor deposition techniques (CVD), necessarily followed by the delicate transfer and patterning of the layered material. In our opinion, the OMIECs hence display a clear advantage in terms of processability.

3. If the additional layers added are lossy, what would be the significance of adding the OMIEC layers onto the metasurface?

We are not exactly sure of which lossy layer the reviewer is referring to in their comment since, besides the substrate and the metallic metasurface, the only additional thin-film deposited on the OMIEC is the iongel. The latter is necessary as a reservoir for the ionic species that enable the electrostatic gating of the OMIEC, and while introducing some losses, these are nonetheless limited – see Figures 1b and 3c. Note that other metadvice technologies based on graphene use similar iongel electrolytes²².

The substrate may also introduce losses, however in this case there is not a strong constraint on the use of different materials less subject to this undesired effect.

4. Additionally, with the flexible metasurface the amplitude and phase modulation is decent, however the transmission amplitudes appear to be low. Is there a possibility of improvement of the design for better amplitudes?

As discussed in the previous comment, the low transmission of the metadvice at rest may be due to the losses introduced by either the substrate or the iongel. While the former can be readily substituted with a better performing material, the latter may require a careful optimization of the thickness to ensure appropriate electrical characteristics while minimizing THz absorption.

Another effect that may influence the maximum transmittance of the metadvice – i.e. dedoped OMIEC at $V_{gate} = 0.6$ V – is the accidental and unwanted oxidation of the mixed conductor, which results in a more conductive OFF state, and thus a lower transmission of the metadvice²³.

5. What's the efficiency of this metasurface?

The optical efficiency of the metasurface can be determined from Supplementary Fig. 5, which presents a plot of the transmission for the metasurface alone, evaluated relative to the substrate. The key metric here is the peak transmission, which is significantly influenced by the specific configuration of the metasurface. The slight variations in resonant frequency arise from the different interactions between the metasurface and the substrates used.

For the fused silica substrate, we measured a transmission of 40% for the gold metasurface and approximately 55% for the silver metasurface, the latter being thicker and thus having lower sheet resistance. On the polyimide substrate, we observed a transmission of roughly 60%.

6. In Supp. Fig. 5 there seems to be a 'stronger' amplitude noted from a different fabrication method using Plastic and Ag. Was it more cost effective to fabricate with quartz instead of plastic, even though there was a higher Q-factor?

Supplementary Fig. 5 indeed shows how the direct-written metasurface fabricated on the plastic substrate is characterized by a higher transmission at resonance with respect to the other two devices made on quartz. We primarily attribute this significant difference in optical absorption to the lower thickness of the polyimide substrate (50 μ m) versus the quartz (1 mm).

We confirm that – at the laboratory scale – quartz is indeed more cost effective than polyimide films. However, our choice to implement the device onto a such substrate was rather determined by the need to establish a robust fabrication protocol on a well-controlled rigid surface before progressing to flexible substrates, which typically require a more careful handling.

References

1. Leitenstorfer, A. *et al.* The 2023 terahertz science and technology roadmap. *J. Phys. Appl. Phys.* **56**, 223001 (2023).
2. Quevedo-Teruel, O. *et al.* Roadmap on metasurfaces. *J. Opt. U. K.* **21**, 073002 (2019).
3. Rivnay, J. *et al.* Organic electrochemical transistors. *Nat. Rev. Mater.* **3**, 17086 (2018).

4. Cea, C. *et al.* Integrated internal ion-gated organic electrochemical transistors for stand-alone conformable bioelectronics. *Nat. Mater.* 1–9 (2023) doi:10.1038/s41563-023-01599-w.
5. Gumyusenge, A., Melianas, A., Keene, S. T. & Salleo, A. Materials Strategies for Organic Neuromorphic Devices. *Annu. Rev. Mater. Res.* **51**, 47–71 (2021).
6. Lloyd-Hughes, J. & Jeon, T.-I. A Review of the Terahertz Conductivity of Bulk and Nano-Materials. *J. Infrared Millim. Terahertz Waves* **33**, 871–925 (2012).
7. Jeon, T.-I., Grischkowsky, D., Mukherjee, A. K. & Menon, R. Electrical and optical characterization of conducting poly-3-methylthiophene film by THz time-domain spectroscopy. *Appl. Phys. Lett.* **79**, 4142–4144 (2001).
8. Unuma, T., Fujii, K., Kishida, H. & Nakamura, A. Terahertz complex conductivities of carriers with partial localization in doped polythiophenes. *Appl. Phys. Lett.* **97**, 033308 (2010).
9. Tsokkou, D., Cavassin, P., Rebetez, G. & Banerji, N. Bipolarons rule the short-range terahertz conductivity in electrochemically doped P3HT. *Mater. Horiz.* **9**, 482–491 (2022).
10. Hegmann, F. A., Ostroverkhova, O. & Cooke, D. G. Probing Organic Semiconductors with Terahertz Pulses. in *Photophysics of Molecular Materials* 367–428 (John Wiley & Sons, Ltd, 2005). doi:10.1002/3527607323.ch7.
11. Quill, T. J. *et al.* Charge Carrier Induced Structural Ordering And Disordering in Organic Mixed Ionic Electronic Conductors. *Adv. Mater.* **36**, 2310157 (2024).
12. Wu, H.-Y. *et al.* Influence of Molecular Weight on the Organic Electrochemical Transistor Performance of Ladder-Type Conjugated Polymers. *Adv. Mater.* **34**, 2106235 (2022).
13. Khim, D. *et al.* Uniaxial Alignment of Conjugated Polymer Films for High-Performance Organic Field-Effect Transistors. *Adv. Mater.* **30**, 1705463 (2018).
14. Natali, D. & Caironi, M. Charge Injection in Solution-Processed Organic Field-Effect Transistors: Physics, Models and Characterization Methods. *Adv. Mater.* **24**, 1357–1387 (2012).
15. Capolino, F., Khajavikhan, M. & Alù, A. Metastructures: From physics to application. *Appl. Phys. Lett.* **120**, 060401 (2022).
16. Ra’Di, Y. & Alù, A. Reconfigurable Metagratings. *ACS Photonics* **5**, 1779–1785 (2018).

17. Zheludev, N. I. & Kivshar, Y. S. From metamaterials to metadevices. *Nat. Mater.* **11**, 917–924 (2012).
18. Venkatesh, S., Lu, X., Saeidi, H. & Sengupta, K. A high-speed programmable and scalable terahertz holographic metasurface based on tiled CMOS chips. *Nat. Electron.* *2020 312* **3**, 785–793 (2020).
19. Paudel, P. R., Skowrons, M., Dahal, D., Radha Krishnan, R. K. & Lüssem, B. The Transient Response of Organic Electrochemical Transistors. *Adv. Theory Simul.* **5**, 2100563 (2022).
20. Piatti, E. *et al.* Charge transport mechanisms in inkjet-printed thin-film transistors based on two-dimensional materials. *Nat. Electron.* **4**, 893–905 (2021).
21. Ning, J. *et al.* Flexible field-effect transistors with a high on/off current ratio based on large-area single-crystal graphene. *Carbon* **163**, 417–424 (2020).
22. Balci, O. *et al.* Electrically switchable metadevices via graphene. *Sci. Adv.* **4**, eaao1749 (2018).
23. Keene, S. T., Melianas, A., van de Burgt, Y. & Salleo, A. Mechanisms for Enhanced State Retention and Stability in Redox-Gated Organic Neuromorphic Devices. *Adv. Electron. Mater.* **5**, 1800686 (2019).

Response to the reviewers:

We thank again all reviewers for their useful comments and questions.

Below our response to the final comments from Reviewer #2:

The authors have provided detailed and interesting responses to the reviewer questions. However, a lot of these great answers then don't translate into the main text. It would help the reader a lot to have key points from the information in the response, in the manuscript.

We thank the Reviewer for their positive assessment of both our work and our response to the various questions emerged from the peer review. We certainly agree that in our Response many topics were discussed to a greater extent relative to the (amended) manuscript, yet we find that such in depth argumentation would likely weaken the readability of the article. We hence welcome the possibility for the readership to access the Peer Review files, as they are made available by Nature Communications.

Also, Figure 2a still needs to be labelled for clarity.

We have again modified the labels on the device schematics to further clarify the image:

The manuscript titled "Tuning direct-written terahertz metadevices with organic mixed ion-electron conductors" presents the usage of organic materials in combination with metasurfaces on both rigid and flexible substrates. A detailed investigation on the work has been presented. As the paper demonstrates potential, I recommend acceptance with revisions. I have some suggestions and questions to the authors to improve the paper.

1. What is the response time of the tuneable metasurface?
2. Can the authors further comment on the advantages of using the organic mixed ion-electron conductors as opposed to other media for tuning metasurfaces such as graphene?
3. If the additional layers added are lossy, what would be the significance of adding the OMIEC layers onto the metasurface?
4. Additionally, with the flexible metasurface the amplitude and phase modulation is decent, however the transmission amplitudes appear to be low. Is there a possibility of improvement of the design for better amplitudes?
5. What's the efficiency of this metasurface?
6. In Supp. Fig. 5 there seems to be a 'stronger' amplitude noted from a different fabrication method using Plastic and Ag. Was it more cost effective to fabricate with quartz instead of plastic, even though there was a higher Q-factor?